# Selectivity of Hydroxamate- and Difluoromethyloxadiazole-Based Inhibitors of Histone Deacetylase 6 In Vitro and in Cells

**DOI:** 10.3390/ijms24054720

**Published:** 2023-03-01

**Authors:** Jakub Ptacek, Ivan Snajdr, Jiri Schimer, Zsofia Kutil, Jana Mikesova, Petra Baranova, Barbora Havlinova, Werner Tueckmantel, Pavel Majer, Alan Kozikowski, Cyril Barinka

**Affiliations:** 1Institute of Biotechnology CAS, BIOCEV, Prumyslova 595, 252 50 Vestec, Czech Republic; 2Institute of Organic Chemistry and Biochemistry of the Academy of Sciences of the Czech Republic, Flemingovo n. 2, 166 10 Prague 6, Czech Republic; 3StarWise Therapeutics LLC, University Research Park, Inc., Madison, WI 53719, USA; 4Department of Medicinal Chemistry and Pharmacognosy, College of Pharmacy, University of Illinois at Chicago, Chicago, IL 60612, USA

**Keywords:** metallohydrolase, histone deacetylase, inhibitor profiling, tubulin/histone acetylation, nanoBRET

## Abstract

Histone deacetylase 6 (HDAC6) is a unique member of the HDAC family of enzymes due to its complex domain organization and cytosolic localization. Experimental data point toward the therapeutic use of HDAC6-selective inhibitors (HDAC6is) for use in both neurological and psychiatric disorders. In this article, we provide side-by-side comparisons of hydroxamate-based HDAC6is frequently used in the field and a novel HDAC6 inhibitor containing the difluoromethyl-1,3,4-oxadiazole function as an alternative zinc-binding group (compound **7**). In vitro isotype selectivity screening uncovered HDAC10 as a primary off-target for the hydroxamate-based HDAC6is, while compound **7** features exquisite 10,000-fold selectivity over all other HDAC isoforms. Complementary cell-based assays using tubulin acetylation as a surrogate readout revealed approximately 100-fold lower apparent potency for all compounds. Finally, the limited selectivity of a number of these HDAC6is is shown to be linked to cytotoxicity in RPMI-8226 cells. Our results clearly show that off-target effects of HDAC6is must be considered before attributing observed physiological readouts solely to HDAC6 inhibition. Moreover, given their unparalleled specificity, the oxadiazole-based inhibitors would best be employed either as research tools in further probing HDAC6 biology or as leads in the development of truly HDAC6-specific compounds in the treatment of human disease states.

## 1. Introduction

Histone deacetylases (HDACs) are a family of proteins responsible for catalyzing the hydrolysis of the acetylated lysine residues of protein substrates, such as histones, to afford the corresponding unsubstituted lysine residue. The action of the HDACs (also referred to as KDACs) is the opposite of the histone acetyltransferases (HATs). There are at present eighteen known mammalian HDACs, which are divided into four classes based on their sequence similarity to yeast homologs: class I (HDAC1, 2, 3, and 8), class IIa (HDAC4, 5, 7, and 9), class IIb (HDAC6 and 10), class III (the sirtuins SIRT1-7), and class IV (HDAC11) [1]. Among these isoforms, HDAC6 has attracted particular attention due to its relative uniqueness within its family. (a) Unlike its related family members, HDAC6 contains two tandem protein deacetylase domains (DD1 and DD2) and has a *N*-terminal nuclear export sequence and a *C*-terminal ubiquitin-binding domain [2,3]. (b) Unlike other HDAC family members distributed in the nucleus, HDAC6 is found primarily within the cytosol and it is responsible for regulating the acetylation state of specific cytosolic proteins. (c) HDAC6 has no apparent role in the post-translational modifications (PTMs) of histone proteins, but instead is involved in regulating the acetylation status of *α*-tubulin, cortactin, heat shock protein 90, HSF-1, and other protein targets. Most importantly, HDAC6 regulation of *α*-tubulin acetylation has sparked considerable interest for a good reason, as the acetylation status of *α*-tubulin affects microtubule mechanical properties and transport of cargos along microtubule tracks by kinesin-1 and dynein motor proteins [4,5,6,7,8]. In addition to facilitating the anterograde transport of new cargo to synaptic zones, acetyl-tubulin also increases the ability of damaged organelles or misfolded proteins to leave synaptic zones [9]. Selective inhibition of HDAC6 leads to neuroprotective effects, and as a consequence, this enzyme has emerged as a particularly attractive therapeutic target by improving mitochondrial movement and axonal transport in conditions such as spinal cord injury, as well as a host of other CNS disorders, including depression [10], stroke [11], Parkinson’s disease [12], Alzheimer’s disease [13,14,15], Charcot–Marie–Tooth disease [16,17], Rett Syndrome [13,18], amyotrophic lateral sclerosis [19], and Fragile-X syndrome [20].

A number of broad-spectrum HDAC inhibitors (HDACis) are in clinical trials as anti-cancer agents due to their ability to improve both histone acetylation and tubulin acetylation, and to promote cell-cycle arrest, differentiation, and apoptosis. Four of these HDACis, namely vorinostat, romidepsin, panobinostat, and belinostat, have been approved by the FDA for applications in cancer therapy. However, significant concerns have emerged that pan-HDACis may be acting too indiscriminately for clinical use beyond oncology, especially with regard to their potential genotoxicity [21,22]. In the last decade, a number of so-called selective HDAC6 inhibitors (HDAC6is) have been disclosed [23] and two of these, rocilinostat (ACY-1215, [24]) and citarinostat (ACY-241), were investigated in clinical trials for different types of cancer [21,22,25,26,27]. Another HDAC6 inhibitor, Tubastatin A, has been used extensively in biological studies in cellular systems and animals, and research with this compound has been considered to further support the possibility to treat certain disease states with HDAC6 inhibitors [28].

Recently, however, there have emerged other HDAC6is that appear to be even more selective than the typical hydroxamic acid-based inhibitors. These newer agents contain an oxadiazole group as the key zinc binding group, and as such, it is believed that these compounds would provide safer alternatives to the typical hydroxamic acid-based inhibitors [29,30]. This notion relates to the fact that the hydroxamic acid group can undergo the Lossen rearrangement, thus leading to an isocyanate that is able to interact with DNA, leading to potential genotoxicity [31].

Thus, in order to better advance the use of HDAC6is for neurological and psychiatric indications, we have undertaken an effort to better characterize some of these newer HDAC6is and to contrast their potency and selectivity with a structurally related hydroxamic acid analog. We believe that this work provides substantial support for the development of novel non-hydroxamic acid-based HDAC6is as potential treatments for a host of disease states, and in cases where the compounds are brain penetrable, for a variety of CNS disorders [31].

## 2. Materials and Methods

### 2.1. HDAC Inhibitors and Substrates

HDAC inhibitors were purchased from the following vendors: ACY-1215 (item no. 21531, Cayman Chemical Company, Ann Arbor, MI, USA), ACY-775 (item no. S0864, Selleckchem, Houston, TX, USA), Tubastatin A (item no. S8049, Selleckchem, Houston, TX USA), Tubacin (item no. S2239, Selleckchem, Houston, TX, USA), Trichostatin A from Streptomyces sp (TSA; item no. BIT1005, Apollo Scientific, Bredbury, Stockport, UK), Nexturastat A (item no. S7473, Selleckchem, Houston, TX, USA). Compounds **6** and **7** were synthesized in-house (see below).

### 2.2. Chemicals

Unless stated otherwise, chemicals were purchased from Sigma-Aldrich, TCI, or Combi-Blocks and were used without further purification. TLC was performed on silica gel 60 F254-coated aluminum sheets (Merck). Products were purified by preparative scale HPLC on a JASCO PU-975 instrument (flow rate 10 mL/min) equipped with a UV-975 UV detector and Waters YMCPACK ODS-AM C18 Prep Column (5 µm, 20 × 250 mm). The purity of compounds was assessed on an analytical JASCO PU-1580 HPLC (flow rate 1 mL/min, invariable gradient from 2 to 100% acetonitrile in 30 min) with a Watrex C18 Analytical Column (5 µm, 250 × 5 mm). 1H and 13 C NMR spectra were measured using Bruker AVANCE III HD 400 MHz, Bruker AVANCE III HD 500 MHz, and Bruker AVANCE III 600 MHz instruments at room temperature, unless noted otherwise. Chemical shifts are given in the δ scale and coupling constants (J) are given in Hz. ESI mass spectra were recorded using a ZQ micromass mass spectrometer (Waters) equipped with an ESCi multimode ion source and controlled by MassLynx software. Low-resolution ESI mass spectra were recorded using a quadrupole orthogonal acceleration time-of-flight tandem mass spectrometer (Q-Tof micro, Waters, Milford, MA, USA) and high-resolution ESI mass spectra using a hybrid FT mass spectrometer combining a linear ion trap MS and Orbitrap mass analyzer (LTQ Orbitrap XL, Thermo Fisher Scientific, Waltham, MA, USA).

#### 2.2.1. *N*-(3-chlorophenyl)methanesulfonamide (**1**)

3-Chloroaniline (1.65 mL, 15.6 mmol, 1.0 equiv.) and pyridine (1.40 mL, 17.3 mmol, 1.1 equiv.) were dissolved in anhydrous DCM (40 mL). Methanesulfonyl chloride (1.46 mL, 18.8 mmol, 1.2 equiv.) was added and the resulting mixture was stirred at room temperature for 2 h under an inert atmosphere. The reaction mixture was washed with 1 M aq. HCl (40 mL), and the organic phase was dried over Na_2_SO_4_. Volatiles were evaporated and the residue was recrystallized from a toluene/cyclohexane mixture (2:1) to afford 2.80 g (87%) of *N*-(3-chlorophenyl)methanesulfonamide (**1**) as pink crystals. The ^1^H NMR spectrum was in agreement with the previously published data [32].

^1^H NMR (401 MHz, CDCl_3_): δ 7.29 (t, *J* = 8.1 Hz, 1H), 7.26 (t, *J* = 2.1 Hz, 1H), 7.19–7.15 (m, 1H), 7.13–7.09 (m, 1H), 6.67 (br s, 1H), and 3.05 (s, 3H).

#### 2.2.2. Methyl 6-(bromomethyl)nicotinate (**2**)

Methyl 6-methylnicotinate (1.40 g, 9.26 mmol, 1.0 equiv.) and *N*-bromosuccinimide (1.81 g, 10.2 mmol, 1.1 equiv.) were dissolved in anhydrous benzene (30 mL). Dibenzoyl peroxide (0.11 g, 0.46 mmol, 0.05 equiv.) was added and the resulting mixture was heated to 80 °C for 18 h under an inert atmosphere. Volatiles were evaporated and the residue was subjected to flash column chromatography (silica gel 60, 70–230 mesh, solvent: cyclohexane/ethyl acetate 4:1) to afford 310 mg (15%) of methyl 6-(bromomethyl)nicotinate (**2**) as a white solid. The ^1^H NMR spectrum was in agreement with the previously published data [33].

^1^H NMR (401 MHz, CDCl_3_): δ 9.17 (dd, *J* = 0.9, 2.2 Hz, 1H), 8.30 (dd, *J* = 2.2, 8.1 Hz, 1H), 7.54 (dd, *J* = 0.9, 8.1 Hz, 1H), 4.58 (s, 2H), and 3.96 (s, 3H).

#### 2.2.3. Methyl 6-((*N*-(3-chlorophenyl)methylsulfonamido)methyl)nicotinate (**3**)

*N*-(3-Chlorophenyl)methanesulfonamide (**1**, 0.20 g, 1.00 mmol, 1.0 equiv.) was dissolved in anhydrous DMF (8 mL) and 60% NaH in mineral oil (48 mg, 1.20 mmol, 1.2 equiv.) was added at 0 °C. After 30 min, compound **2** (0.23 g, 1.00 mmol, 1.0 equiv.) was added and the reaction mixture was allowed to warm to room temperature and stirred overnight. After 16 h, volatiles were evaporated and the residue was subjected to flash column chromatography (silica gel 60, 70–230 mesh, solvent: dichloromethane/methanol 50:1) to afford 330 mg (93%) of methyl 6-((*N*-(3-chlorophenyl)methanesulfonamido)methyl)nicotinate (**3**) as a light yellow solid.

^1^H NMR (401 MHz, CDCl_3_): δ 9.14–9.10 (m, 1H), 8.31–8.22 (m, 1H), 7.50 (d, *J* = 8.20 Hz, 1H), 7.45–7.40 (m, 1H), 7.32–7.21 (m, 3H), 5.07 (s, 2H), 3.94 (s, 3H), and 3.06 (s, 3H).

^13^C NMR (101 MHz, CDCl_3_): δ 165.6, 160.7, 150.6, 140.8, 138.2, 135.1, 130.5, 128.4, 128.1, 126.1, 125.3, 122.0, 56.4, 52.6, and 38.8.

ESI MS: *m/z* 355.1 ([M + H]^+^).

HR ESI MS: calculated *m/z* for C_15_H_16_O_4_N_2_ClS 355.05138; found *m/z* 355.05133.

#### 2.2.4. *N*-(3-chlorophenyl)-*N*-((5-(hydrazinecarbonyl)pyridin-2-yl)methyl)methanesulfonamide (**4**)

Compound **3** (0.35 g, 1.00 mmol, 1.0 equiv.) was dissolved in ethanol (10 mL) and hydrazine hydrate (50% aq. solution, 0.53 mL, 10 equiv.) was added. The resulting mixture was stirred at 80 °C for 16 h. Volatiles were then removed under reduced pressure and the product was purified using the preparative HPLC with a gradient of 0–50% acetonitrile in H_2_O (with 0.1% TFA). The fractions containing the desired product were collected and lyophilized to afford 320 mg (91%) of *N*-(3-chlorophenyl)-*N*-((5-(hydrazinecarbonyl)pyridin-2-yl)methyl)methanesulfonamide (**4**) as a white solid.

^1^H NMR (401 MHz, CD_3_OD): δ 8.92 (dd, *J* = 0.8, 2.2 Hz, 1H), 8.22 (dd, *J* = 2.3, 8.2 Hz, 1H), 7.67 (d, *J* = 8.2 Hz, 1H), 7.51 (t, *J* = 1.9 Hz, 1H), 7.43–7.38 (m, 1H), 7.33 (t, *J* = 7.9 Hz, 1H), 7.31–7.26 (m, 1H), 5.11 (s, 2H), and 3.11 (s, 3H).

^13^C NMR (101 MHz, CD_3_OD): δ 166.6, 162.3, 149.3, 142.4, 137.8, 135.7, 131.6, 129.6, 129.1, 127.6, 127.4, 123.9, 56.8, and 38.1.

ESI MS: *m/z* 355.1 ([M + H]^+^).

HR ESI MS: calculated *m/z* for C_14_H_16_O_3_N_4_ClS 355.06262; found *m/z* 355.06241.

#### 2.2.5. *N*-(3-Chlorophenyl)-*N*-((5-(2-(2,2-difluoroacetyl)hydrazine-1-carbonyl)pyridin-2-yl)methyl)methanesulfonamide (**5**)

Compound **4** (36 mg, 0.10 mmol, 1.0 equiv.) and trimethylamine (42 µL, 0.30 mmol, 3 equiv.) were dissolved in anhydrous DCM (2 mL) at 0 °C and difluoroacetic anhydride (12.4 µL, 0.10 mmol, 1.0 equiv.) was added. The resulting mixture was stirred at 0 °C for 1 h. Volatiles were then removed under reduced pressure and the product was purified by preparative HPLC with a gradient of 15–50% acetonitrile in H_2_O (with 0.1% TFA). The fractions containing the desired product were collected and lyophilized to afford 37 mg (85%) of *N*-(3-chlorophenyl)-*N*-((5-(2-(2,2-difluoroacetyl)hydrazine-1-carbonyl)pyridin-2-yl)methyl)methanesulfonamide (**5**) as a white solid.

^1^H NMR (401 MHz, CD_3_OD): δ 8.93 (d, *J* = 1.6 Hz, 1H), 8.25 (dd, *J* = 2.3, 8.2 Hz, 1H), 7.68 (d, *J* = 8.2 Hz, 1H), 7.52 (t, *J* = 1.8 Hz, 1H), 7.4–7.38 (m, 1H), 7.32–7.27 (m, 1H), 7.30 (dt, *J* = 1.7, 7.9 Hz, 1H), 6.23 (t, *J* = 53.2 Hz, 1H), 5.12 (s, 2H), and 3.11 (s, 3H).

^13^C NMR (101 MHz, CD_3_OD): δ 166.5, 164.1(t, *J*_C,F_ = 25.6 Hz), 161.5, 149.0, 142.3, 138.1, 135.7, 131.6, 129.6, 129.2, 128.7, 127.6, 124.0, 109.8 (t, *J*_C,F_ = 249.0 Hz), 56.7, and 38.2.

ESI MS: *m/z* 433.1 ([M + H]^+^).

HR ESI MS: calculated *m/z* for C_16_H_16_O_4_N_4_ClF_2_S 433.05434; found *m/z* 433.05401.

#### 2.2.6. 6-((*N*-(3-Chlorophenyl)methylsulfonamido)methyl)-*N*-hydroxynicotinamide (**6**)

Hydroxylamine hydrochloride (116 mg, 1.67 mmol) H was dissolved in MeOH (1 mL) at 0 °C and a solution of potassium hydroxide (0.19 g, 3.33 mmol) in MeOH (1 mL) was added. The resulting mixture was stirred at 0 °C for 5 min, then centrifuged, and 0.5 mL of this solution was added to compound **3** (50 mg, 0.04 mmol, 1.0 equiv.) in MeOH (1 mL) at 0 °C. The resulting mixture was stirred at room temperature for 16 h. A saturated aqueous solution of ammonium chloride (0.5 mL) was added and the reaction mixture was stirred for 30 min. Volatiles were then removed under reduced pressure and the product was purified by preparative HPLC with a gradient of 15–50% acetonitrile in H_2_O (with 0.1% TFA). The fractions containing the desired product were collected and lyophilized to afford 12 mg (24%) of 6-((*N*-(3-chlorophenyl)methylsulfonamido)methyl)-*N*-hydroxynicotinamide (**6**) as a white solid.

^1^H NMR (600 MHz, DMSO-d_6_): δ 11.32 (br s, 1H), 8.78 (d, *J* = 2.1 Hz, 1H), 8.05 (dd, *J* = 2.2, 8.1 Hz, 1H), 7.58 (t, *J* = 2.0 Hz, 1H), 7.53 (d, *J* = 8.2 Hz, 1H), 7.45–7.41 (m, 1H), 7.38 (t, *J* = 7.9 Hz, 1H), 7.36–7.32 (m, 1H), 5.04 (s, 2H), and 3.18 (s, 3H).

^13^C NMR (101 MHz, DMSO-d_6_): δ 159.1, 158.0, 147.4, 141.2, 135.6, 133.2, 130.6, 127.7, 127.5, 127.4, 126.6, 121.7, 55.2, and 37.9.

ESI MS: *m/z* 354.0 ([M − H]^+^).

HR ESI MS: calculated *m/z* for C_14_H_13_O_4_N_3_ClS 354.03208; found *m/z* 354.03206.

#### 2.2.7. *N*-(3-chlorophenyl)-*N*-((5-(5-(difluoromethyl)-1,3,4-oxadiazol-2-yl)pyridin-2-yl)methyl)methanesulfonamide (7)

Compound **5** (24 mg, 0.05 mmol, 1.0 equiv.) and methyl *N*-(triethylammoniumsulfonyl)carbamate (Burgess reagent, 39 mg, 0.16 mmol, 3 equiv.) were dissolved in anhydrous THF (0.5 mL) at room temperature. The resulting mixture was stirred at 50 °C for 16 h. Volatiles were then removed under reduced pressure and the product was purified by preparative HPLC with a gradient of 20–60% acetonitrile in H_2_O (with 0.1% TFA). The fractions containing the desired product were collected and lyophilized to afford 18 mg (78%) of *N*-(3-chlorophenyl)-*N*-((5-(5-(difluoromethyl)-1,3,4-oxadiazol-2-yl)pyridin-2-yl)methyl)methanesulfonamide (**7**) as a white solid.

^1^H NMR (401 MHz, CDCl_3_): δ 9.24 (d, *J* = 1.6 Hz, 1H), 8.37 (dd, *J* = 2.2, 8.2 Hz, 1H), 7.64 (d, *J* = 8.2 Hz, 1H), 7.47–7.42 (m, 1H), 7.35–7.23 (m, 3H), 6.93 (t, *J* = 51.6 Hz, 1H), 5.11 (s, 2H), and 3.07 (s, 3H).

^13^C NMR (101 MHz, CDCl_3_): δ 164.1, 160.9, 158.7(t, *J*_C,F_ = 29.0 Hz),, 148.0, 140.7, 135.7, 135.2, 130.6, 128.5, 128.1, 126.0, 122.7, 118.5, 105.8 (t, *J*_C,F_ = 241.0 Hz), 56.4, and 38.7.

ESI MS: *m/z* 415.0 ([M + H]^+^).

HR ESI MS: calculated *m/z* for C_19_H_26_O_5_N_2_Na 415.04377; found *m/z* 415.04372.

## 3. Cloning of Expression Plasmids

Clones encoding human HDACs were obtained from different sources, as summarized in Appendix A Appendix A. Coding sequences were amplified by two-step PCR using specific sets of primers, and the Gateway cloning technology (Invitrogen, Carlsbad, CA, USA) was used to obtain individual HDAC expression plasmids as described previously [34]. Briefly, the amplified genes were inserted into the pDONR221 donor plasmid using the BP recombination mix. Expression plasmids were constructed by subsequent LR recombination between the donor plasmid and the pMM322 destination plasmid bearing the N-terminal Strep-FLAG-HALO purification tags (Appendix A).

### 3.1. Heterologous Expression and Purification of Recombinant HDACs

Large-scale expression of human HDACs (and truncated HDAC5) was carried out essentially as described previously [34]. Briefly, HEK293/T17 cells were transiently transfected using linear polyethylene imine (PEI, Polysciences Inc., Warrington, PA, USA). Cells were harvested three days post-transfection and cell pellets were resuspended in a lysis buffer. Cells were disrupted by sonication and cell lysates were cleared by centrifugation at 40,000× *g* for 30 min. The soluble protein fraction was used for subsequent purification exploiting Streptactin affinity chromatography followed by size-exclusion chromatography. The purity of recombinant proteins was monitored by SDS-PAGE, and final protein preparations were concentrated to the desired concentration, aliquoted, flash-frozen in liquid nitrogen, and stored at −80 °C.

### 3.2. Deacetylation Assays In Vitro

Customized assays were used for individual HDAC isoforms to reflect their distinct substrate specificity. All reactions were carried out in the activity buffer comprising 50 mM HEPES, 140 mM NaCl, 10 mM KCl, pH 7.4 supplemented with 1 mg/mL bovine serum albumin (BSA) and 1 mM tris(2-carboxyethyl) phosphine (TCEP).

HDACs 1–9, 11: Assays were performed essentially as described previously [35]. Briefly, the Ac-GAK-Ac-AMC substrate was used to determine the activity of HDACs 1, 2, 3, and 6; and the Boc-Lys(TFA)-AMC substrate was used for HDACs 4, 5, 7, 8, 9, and 11 (both substrates Bachem, Bubendorf, Switzerland). HDAC at the optimized concentration was preincubated with a dilution series of the inhibitor to be tested in 384-well plates (volume 30 µL, Corning, NY, USA) at 37 °C for 15 min. Upon substrate addition (10 µL; 10 µM final concentration), the reaction was allowed to proceed for an additional 30 min and was terminated by the addition of 25 µL of trypsin solution (4 mg/mL stock solution). Released aminometyl coumarine was quantified using a CLARIOstar plate reader (BMG Labtech GmbH, Ortenberg, Germany) at λ_ex_/λ_em_ = 365/440 nm.

HDAC10: Recombinant human HDAC10 (0.5 nM) in the reaction buffer was preincubated with a 3-fold dilution series of a tested inhibitor at 37 °C for 15 min and the reaction initiated by the addition of a substrate (N8-acetylspermidine labeled with fluorescein; 10 µM final concentration) in a total volume of 50 µL. Following 30 min incubation, the reaction was terminated by the addition of 5 µL of 0.5% acetic acid and centrifuged at 2000× *g* at RT for 15 min to remove precipitated BSA. Reaction mixtures were analyzed by RP-HPLC with a Kinetex 2.6 μm XB-C18 100 Å column with a fluorescence detector set to λ_EX_/λ_EM_ = 492/516.

HDAC inhibition was calculated using corresponding non-inhibited reactions as a control. Inhibition data were fitted in the GraphPad Prism software (version 8.0.1, GraphPad Software, San Diego, CA, USA) using nonlinear regression analysis.

### 3.3. MTT Viability Assay

RPMI-8226 lymphoblasts (ATTC #CCL-155) were used to assess the cytotoxicity of tested compounds. Cells were diluted in an RPMI-1640 medium supplemented with 10% FBS to 2.5 × 10^5^ cells/mL and 90 µL of the cell suspension seeded in a 96-well plate. Dilution series of tested inhibitors were prepared as 10x stock solutions in PBS and 10 μL was added to cells and incubated under a 5% CO_2_ atmosphere at 37 °C, for 2 days. Following incubation, 10 μL of MTT reagent (3-(4,5-dimethylthiazol-2-yl)-2,5-diphenyltetrazolium bromide, 5 mg/mL in PBS, Sigma Aldrich) was added to each well and incubated for 90 min, 37 °C, 5% CO_2_. The crystals of metabolized MTT formazan products were dissolved by the addition of 100 μL solubilization solution (40% dimethylformamide, 2% acetic acid, 16% SDS, pH 4.7) and shook at 22 °C for 4 h. Absorbance at 570 nm was measured using CLARIOstar plate reader (BMG Labtech, Ortenberg, Germany) and data were plotted using the GraphPad Prism software.

### 3.4. HDAC1/HDAC6 Engagement Assay

The NanoBRET Target Engagement Intracellular HDAC Assay kit (Promega, Madison, WI, USA) was employed, and the assay was carried out according to the manufacturer’s protocol. Briefly, HEK293/T17 cells were transfected with plasmids carrying nanoLuc-HDAC fusion genes (provided in the kit) using lipofectamine transfection reagent (Invitrogen, Waltham, MA, USA), and cell lines stably expressing the fusion proteins were selected using corresponding antibiotics. For each experiment, cells were expanded and then trypsinized and diluted in the Opti-MEM I reduced serum medium without phenol red (Thermo Scientific) to 200,000 cells/mL. The reaction was carried out on a white non-binding surface 384-well plates (cat. #3574, Corning Inc., Corning, NY, USA), where 34 μL of the cell suspension was mixed with 2 μL of 20× the NanoBRET tracer (final concentration 0.5 μM). Four µL of a 10× dilution series of a test compound was then added, mixed, and further incubated at 37 °C for 2 h. Following incubation, 20 μL of the Nano-Glo substrate/extracellular inhibitor mixture was added and incubated at 22 °C for 2 min, and luminescence was quantified using a CLARIOstar plate reader (BMG Labtech GmbH, Ortenberg, Germany). Donor and acceptor emission wavelengths (460/70 nm and 656/88 nm, respectively) were recorded, and the BRET ratio was calculated using the following equation: [(Acceptor_sample_/Donor_sample_) − (Acceptor_no-tracer control_/Donor_no-tracer control_)] × 1000.

### 3.5. IC_50_ Determination in Cells

Tubulin acetylation levels were used as a surrogate marker to determine inhibitory potency in cells using the protocol described previously [36]. RPMI-8226 lymphoblasts (10^6^ cells/mL) in an RPMI-1640 medium supplemented with 10% FBS were incubated in 96-well plates (90 µL). A dilution series of 10 µL of stock inhibitor solutions in PBS (10 µL) was added to cells and incubated at 37 °C for 6 h. The plate was centrifuged at 500× *g* at 22 °C for 5 min, the supernatant discarded, and cells resuspended in 80 µL of lysis buffer (20 mM Tris, 4 M urea, 5 mM MgCl_2_, 0.5% Triton X-100, pH 8.2 supplemented with protease inhibitors (cOmplete EDTA-free protease inhibitors cocktail, Roche, Basel, Switzerland) and benzonase (2U/mL, Merck, Darmstadt, Germany). Following the 5 min incubation at 22 °C, a 4× SDS-PAGE sample buffer was added and heated to 95 °C for 5 min. Samples (10^4^ cells/lane) were separated by SDS-PAGE and levels of acetylated tubulin (Ac-tub) were determined by quantitative Western blotting.

### 3.6. Quantitative Western Blotting

Samples separated by SDS-PAGE were transferred onto a PVDF membrane using a Trans-Blot Turbo RTA Mini PVDF Transfer kit (Bio-Rad, Hercules, CA, USA). The membrane was blocked with 5% BSA in TBS (50 mM Tris-HCl, 150 mM NaCl, pH 7.4) and incubated with the primary antibody mix in 5% BSA in TBS at 4 °C overnight. Antibodies used were anti-alpha tubulin (1:4000 dilution, rabbit, #Ab18251, Abcam, Cambridge, UK), anti-acetylated tubulin (1:2500 dilution, mouse, #T7451, Sigma-Aldrich, St. Louis, MO, USA), and anti-acetylated histone H3 (Lys9/Lys14) (1:2000 dilution, rabbit, #9677, Cell Signaling, Danvers, MA, USA). The next day, the membrane was washed in TBS + 0.2% Tween 20, and further incubated in a mix of secondary antibodies comprising Alexa Fluor 568-donkey anti-rabbit IgG (0.4 μg/mL, #A10042, Invitrogen, Waltham, MA, USA) and Alexa Fluor 488-goat anti-mouse IgG (0.4 μg/mL, #A11029, Invitrogen, Waltham, MA, USA) at 22 °C for 1 h. Bands were visualized using a Typhoon FLA9500 fluorescence imager (GE Healthcare Bio-Sciences, Little Chalfont, UK) and signal intensities were quantified using Quantity One 1-D Analysis Software (Bio-Rad, Hercules, CA, USA). Signals of Ac-tubulin were normalized to the total tubulin load and the absolute amount of acetylation was determined according to purified tubulin standards with known acetylation levels.

### 3.7. Correlation Analysis

EC_50_ values from the viability MTT assay were plotted against nanoBRET target engagement assay EC_50_s and the correlation analysis was performed using the Spearman nonparametric correlation function in GraphPad Prism software (version 8.0.1, GraphPad Software, San Diego, CA, USA). The resulting correlation coefficient (r) ranges from −1 (perfect negative correlation) through 0 (no correlation) to 1 (perfect positive correlation). The *p*-value defines the probability that a random data set would provide the same (or better) correlation coefficient than observed in the experiment.

## 4. Results and Discussion

### 4.1. Selectivity of Hydroxamate-Based HDAC6-Specific Inhibitors (HDAC6is) Is Limited In Vitro

While in vitro potency and selectivity of HDAC6is have been reported in the literature [10,24,37,38,39], the data are often incomplete, and the use of different experimental protocols makes their comparison challenging. Therefore, we selected five HDAC6is widely used in the field, together with TSA as a representative of a pan-HDAC inhibitor, and first compared their potency against a panel of human zinc-dependent HDACs using standard in vitro deacetylation assays (Figure 1 and Appendix A Appendix A).

IC_50_s of the inhibitors range from low nanomolar to over 50 μM (the high limit of our assays) values. As expected, all compounds inhibit HDAC6 with high potency in the low nanomolar range, with TSA and Tubacin having the lowest (0.4 nM) and the highest (36 nM) IC_50_s, respectively. Interestingly, all compounds, with the exception of Tubacin, also inhibit HDAC10 very potently, and HDAC10 is thus the prime off-target isoform of these HDAC6-selective inhibitors. It should be also noted that data evaluating inhibitor potency against human HDAC10 are often missing and have only been reported frequently in recent years [40,41,42,43] upon the identification of spermidine as the HDAC10 physiologic substrate, which allowed for the development of more reliable and efficient screening assays [44].

Off-targeting HDAC10 by HDAC6is may not come as a surprise, as HDAC6 and HDAC10 both belong to the same class IIb HDAC subfamily and their catalytic domains share high structural homology. At the same time, recent studies pointed toward structural features that might be exploited for the design of compounds discriminating between HDAC6 and HDAC10 [40,42,43,45]. Specific for HDAC10 is a 3_10_ “P(E,A)CE” helix sterically constricting the active site and more importantly the negatively charged gatekeeping glutamate 274. The latter confers selectivity for cationic polyamine substrates and can be selectively engaged by cap functions of inhibitors comprising complementary positively charged groups [43,45,46]. More extensive SAR studies will be required to evaluate whether the presence of the difluoromethyl-1,3,4-oxadiazole zinc-binding group is also one of the critical factors contributing toward HDAC6 selectivity, as alluded in this report.

In addition to HDAC10, class I HDACs (HDAC 1–3, 8) are also targeted by HDAC6is, albeit with lower affinity (Figure 1). In fact, the ratio of HDAC1/HDAC6 inhibition potency is often used in the literature to define the selectivity index (SI), i.e., specificity, of HDAC6is (Table 1) [24,37,38,39]. The focus on class I HDACs is quite understandable, as inhibition of these isoforms can be linked to the undesired cytotoxicity of HDAC6is, which limits their potential therapeutic use in the treatment of chronic diseases.

When compared to the literature data, the IC_50_ values for the known inhibitors against HDAC6 are analogous to the results reported herein (Table 1) [10,24,37,38,39]. Interestingly though, potency against HDAC1 (and thus the selectivity index) differs more widely, with IC_50_ values often being up to 10-fold higher in prior reports (Table 1) [10,24,37,38,39]. These discrepancies most likely result from the use of different in vitro assay conditions. While in this report we use the Ac-GA[AcK]-AMC (GAK) substrate at 10 µM concentration as a substrate, many labs use the RHK[AcK]-AMC (RHKK) peptide at 50 µM concentration in their assays [47,48]. Consequently, we call attention to the fact that Michaelis constants for GAK and RHKK substrates are 9.1 µM and 3.7 µM, respectively. The use of high 50 µM concentrations of RHKK in the assays results in a corresponding increase in the IC_50_ values due to the competition between an inhibitor and the substrate in the HDAC active site. Understandably, reporting and comparing inhibitor potency in the form of (apparent) inhibition constants (*K_i_* or *K_app_*) would bring more consistency and allow direct comparison of data from different laboratories, though such reporting is yet to be implemented in the field.

### 4.2. The Difluoromethyl-1,3,4-Oxadiazole Zinc-Binding Group Imparts High HDAC6 Selectivity

The hydroxamate zinc-binding group (ZBG) imparts high potency to the HDACis, yet it can be associated with off-target inhibition as well as poor pharmacokinetics and possible mutagenicity [49]. For example, hydroxamic acids are susceptible to metabolism by hydrolysis to form carboxylic acids [50] by a reduction in the N-O bond to form carboxamides [51] and by conjugation, e.g., glucuronidation [52]. Neither carboxylic acids nor carboxamides exhibit strong complexation to zinc and are, therefore, typically inactive or poorly active. Glucuronidation turns the OH part of the hydroxamic acid function into a leaving group and thereby activates the N-O bond toward a Lossen rearrangement, with the resulting highly reactive isocyanate being responsible for mutagenic activity [53]. Furthermore, many metalloenzymes, including iron–sulfur cluster proteins, and most prominently the acyl-CoA hydrolase MBLAC2, have been reported to be common hydroxamate-based HDACi off-targets [28]. Therefore, different ZBGs have been investigated as hydroxamate substitutes including mercaptoamides, thiols, hydrazides, and substituted 1,3,4-oxadiazoles [54,55]. Interestingly, while the latter ZBG is widely reported in patents [30,56,57], this group is much less represented in research manuscripts [30,58,59]. Compared to the hydroxamate function, 1,3,4-oxadiazoles do not appear to constitute much of a metabolic liability. At the same time, however, oxidative ring opening to form a 1,2-diacylhydrazine has been observed and is believed to proceed through cytochrome P450-mediated epoxidation of one of the ring’s C=N double bonds [60]. The exact nature of degradation routes and degradation products for 1,3,4-oxadiazole-containing HDACis has not been evaluated in vivo and will, among other factors, depend on the chemistry of the linker/capping groups and will require pharmacokinetic evaluation specifically targeted to every compound tested.

To assess isoform selectivity of difluoromethyl-1,3,4-oxadiazole-based HDAC6is, we synthesized a representative HDAC6i described in the patent literature (compound **7**; [56]) and further prepared a corresponding hydroxamate analog (compound **6**; Figure 1). In vitro potency and selectivity of both compounds were evaluated against a panel of zinc-dependent human HDACs as described above. The hydroxamate-based compound **6** has the highest potency of all tested compounds, with IC_50_ = 0.35 nM against HDAC6. At the same time, it also inhibits class I and HDAC10 isoforms in the nanomolar range, with a modest selectivity index of 141 and 195 for HDAC1 and HDAC10, respectively. In comparison, while being slightly less potent against HDAC6 (IC_50_ = 2.1 nM), the 1,3,4-oxadiazole analog **7** exhibits exquisite > 10,000-fold selectivity for HDAC6 over all other HDAC isoforms (Figure 1, Table 1 and Appendix A Appendix A). Clearly, in terms of isoform selectivity, compound **7** outperforms all of the other HDAC6is evaluated here.

### 4.3. Cellular Potency of HDAC6is Is on Average 100-Fold Lower Compared to In Vitro Data

In vitro profiling, as described above, is typically used as the first step in the preclinical development of enzyme inhibitors, and this assay is then followed by cellular readouts, and later by in vivo experiments. To compare the specificity and potency of the tested inhibitors against HDAC1 and six isoforms in a cellular environment, we employed the nanoBRET (BRET = bioluminescence resonance energy transfer) target engagement assay [61,62]. To this end, we first established HEK293T/17 cell lines stably expressing HDAC 1 and six isoforms fused to a luciferase reporter (HDAC-nanoLuc). Then, the cell lines were treated with a dilution series of the studied HDAC6is to directly determine their EC_50_ values in the intracellular environment (Figure 2).

For HDAC6, the cellular inhibitory potency is approximately 100- to 1000-fold lower compared to in vitro data, ranging from 0.16 µM to 17.5 µM for **7** and Tubacin, respectively. Similarly, micromolar EC_50_ values were determined at HDAC1, with the exception of TSA, where the EC_50_ value was 40 nM, which is in line with TSA specificity for class I HDACs. We note that the EC_50_s against HDAC1 could not be calculated for Tub A, ACY-775, and compound **7**, as these are higher than the highest inhibitor concentration used (40 µM). These findings suggest that Tub A, ACY-775, and especially compound **7**, are the best candidates for cell-based assays where high HDAC6-selectivity is required. Furthermore, the said inhibitors are best used at low (1–10 µM) micromolar concentrations to limit off-target effects. Interestingly, the HDAC6/HDAC1 selectivity index is much lower in the nanoBRET assay compared to the in vitro data, and selectivity is even reversed for ACY-1215 and Tubacin (Figure 2). Overall, our nanoBRET data can serve as a cautionary tale of oversimplified reliance on in vitro specificity data when HDAC6is are used in cells/in vivo. Furthermore, the physiological effects of cell treatment with “selective” HDAC6is are best interpreted and assigned with caution, as these results may derive from a more “complex” inhibition of several HDAC isoforms rather than being simply ascribed to HDAC6 function(s).

Tubulin and histones are the most prominent substrates of HDAC6 and class I HDACs, respectively, and tubulin/histone acetylation serves as a surrogate readout when determining HDACi potency in cells. To extend and complement our nanoBRET cellular experiments, we further evaluated HDAC6i potency and selectivity by quantitative Western blotting. To this end, RPMI-8226 lymphoblasts were treated with a dilution series of HDAC6is for six hours and the tubulin/histone acetylation levels were quantified using specific antibodies (Figure 3 and Appendix A). For HDAC6, the EC_50_ values are all in the high nanomolar to low micromolar ranging from 0.14 μM to 1.7 μM for TSA and Tubacin, respectively, confirming an approximately 100- to 1000-fold decrease in potency as compared to the in vitro data, with Tubacin being consistently the HDAC6i with the lowest inhibitory potency. On the other hand, the correlation between nanoBRET and Ac-tubulin is, perhaps surprisingly, relatively weak, although the modest number of inhibitors tested limits the generalization of these findings (Figure 3D).

While marked (1000-fold) differences were determined in EC_50_ values against HDAC1 using the nanoBRET assay, data from Western blotting, where histone H3 acetylated at positions K9/K14 is used as a surrogate readout for class I HDAC inhibition, are somewhat less distinctive (Figure 3C). In line with the in vitro and nanoBRET data, general trends are maintained, with TSA and ACY-1215 being the most potent class I HDAC inhibitors, while compound **7** does not increase H3 acetylation even at the highest concentrations, thus confirming its superior HDAC6 selectivity. At the same time, the 10-fold difference between the EC_50_ for NextA and TubA observed by HDAC1-targeted nanoBRET is not clearly recaptured using Western blots. We believe that several interconnected issues need to be considered when looking into the class I HDAC inhibition by Western blotting. First, technically, it appears that the specificity and the dynamic range of the Ac-K9/K14-H3 mAb are limited and that other acetylation markers/antibodies might be better suited for this purpose. Second, histone acetylation is regulated by more than one HDAC isoform, and thus a simple correlation between HDAC1-targeted nanoBRET and histone acetylation status is unlikely, and at a minimum, both HDAC2 and HDAC3 should be also considered during data analysis. Finally, class I HDAC can be incorporated into different physiological complexes, and HDAC inhibition profiles are dependent on the particular complex and not just the defined HDAC component [63]. Overall, while histone acetylation status can serve as a qualitative indicator of class I HDAC inhibition, these quantitative readouts would appear to have limited utility.

### 4.4. Cellular Toxicity of HDACis Correlates with Class I HDAC Off-Targeting

Our in vitro and nanoBRET inhibition data show that the specificity of the tested HDAC6is, with the exception of compound **7**, is not restricted to HDAC6 only, but marked inhibition of other enzyme isoforms is observed. Of special importance is the off-target inhibition of class I HDACs, as this is associated with the cytotoxicity of a given compound [64]. While class I HDAC inhibition and associated cytotoxicity are desired in the oncology field, it would be unfavorable in chronic diseases, such as neurodegeneration, where long-term targeting of HDAC6 is required [65]. In the final set of experiments, we thus evaluated the direct cytotoxicity of studied HDAC6is using RPMI-8226 lymphoblasts as a model. Cells were treated with dilution series of HDAC6is for 48 h and the cell viability was determined using the MTT assay. As expected, and in excellent agreement with all assays, TSA turned out to be the most cytotoxic compound, with a calculated EC_50_ value of 0.24 μM. Additionally, ACY-1215, NextA, and Tubacin, which also inhibit class I HDACs potently, show cytotoxic EC_50_ values of 2.9, 2.8, and 6.5, μM, respectively. On the other hand, both TubA and, most notably, compound **7,** had less pronounced effects on cell viability at high inhibitor concentrations (Figure 4). Finally, we observed a strong correlation between inhibitor cytotoxicity and in vitro (nanoBRET) HDAC1 inhibition (correlation coefficient r = 0.937, *p*-value = 0.0048), while no clear correlation exists between cytotoxicity and HDAC6 inhibition (r = −0.179, *p* = 0.71; Figure 4B). Overall, the findings reported herein suggest that the MTT (or other cell viability) assay might be preferred over Ac-H3 quantification for determining to what extent HDAC6is possesses possible off-target activity at class I HDACs, as these viability assays are more quantitative, less demanding, and easily adaptable to high-throughput formats.

## 5. Conclusions

HDAC6-specific inhibitors represent tools instrumental for deciphering the physiological functions of this isoform that could be further used for the therapy of various human disease states. At the same time, critical assessment of physiological readouts is warranted, as hydroxamate-based HDAC6is used in the field can inhibit other HDAC isoforms, with HDAC10 being particularly targeted. HDAC6is containing the difluoromethyl-1,3,4-oxadiazole function as an alternative zinc-binding group can offer unparalleled selectivity over all other HDAC isoforms and should thus be considered as preferred alternatives for the development of HDACis in the future.

## Data Availability

Not applicable.

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
