# Peer review of "Selectivity of Hydroxamate- and Difluoromethyloxadiazole-Based Inhibitors of Histone Deacetylase 6 In Vitro and in Cells"

_ijms, 2023, doi:10.3390/ijms24054720_

Round 1

Reviewer 1 Report

The authors compared the selectivity of HDAC6 inhibitors using the in-vitro deacetylation, nanoBRET, WB, and MTT assays, and identified the oxadiazole-based compound 7 with high selectivity over other HDACs. This work may thus facilitate the discovery of HDAC6 specific inhibitors in the treatment of human diseases, such as neurological/psychiatric disorders.

 Minor points:

Compounds 6 and 7 are referred in Fig. 1 and Table 1, but its citation is in line 393. It lowers the readability.

Table 1, unit of IC50 is missing.

Figure 2B, inhibitor concentration should be uM?

Figure 2C, the abbreviations of SI, HD6, HD1 should be included either in the main text or in the legend.

L501-503, quantification of the correlation would be preferred.

Reviewer 2 Report

The paper by Jakub Ptacek et al., is very well written, with not so many corrections which should be taken into account during the revision.

1. Could you add a brief explanation to the Introduction, what is the principle from the 3D structural point of view for selectivity of inhibitors of HDAC6, compared to HDAC10, which belongs to the class IIb too?

2. In Methods the formulas of the compounds should be organized as a figure (Scheme 1, which better suits to Methods).

3. The figure 1 should be slightly modified. Currently Fig 1A occupies most of the space, while Fig 1B is more important.
It's better to rearrange the Fig 1A and show a larger image of transposed Fig. 1B (show HDACs as columns).

4. Tables and figures should be self-explanatory. Currently very little is written in the description of Table 1. Please also add dimensions and explain abbreviations where appropriate.

5. lines 501-503. The mentioned correlations could be evaluated statistically. Please add R and p-values for such correlations and compare them in the text here.  

6. Surprisingly there is no discussion. A brief discussion of the presence of other, non-HDAC off-targets (possibly among zinc-dependent enzymes) can be added. Also a discussion of in vivo degradation routs and degradation products of the HDAC inhibitors (compound 7 vs other ones) is needed. Clinical effects of the inhibitors can be also discussed here.
